# Deep mutagenesis scanning using whole trimeric SARS-CoV-2 spike highlights the importance of NTD-RBD interactions in determining spike phenotype

Ruthiran Kugathasan[1], Ksenia Sukhova[1], Maya Moshe[1], Paul Kellam[1,2], Wendy Barclay[1] *

1 Department of Infectious Diseases, Imperial College London, London, United Kingdom, 2 RQ Biotechnology Ltd, London, United Kingdom

* w.barclay@imperial.ac.uk

**Data Availability Statement:** The authors confirm that all data underlying the findings are fully available without restriction. The raw enrichment

## Abstract

New variants of SARS-CoV-2 are continually emerging with mutations in spike associated with increased transmissibility and immune escape. Phenotypic maps can inform the prediction of concerning mutations from genomic surveillance, however most of these maps currently derive from studies using monomeric RBD, while spike is trimeric, and contains additional domains. These maps may fail to reflect interdomain interactions in the prediction of phenotypes. To try to improve on this, we developed a platform for deep mutational scanning using whole trimeric spike. We confirmed a previously reported epistatic effect within the RBD affecting ACE2 binding, that highlights the importance of updating the base spike sequence for future mutational scanning studies. Using post vaccine sera, we found that the immune response of vaccinated individuals was highly focused on one or two epitopes in the RBD and that single point mutations at these positions can account for most of the immune escape mediated by the Omicron BA.1 RBD. However, unexpectedly we found that the BA.1 RBD alone does not account for the high level of antigenic escape by BA.1 spike. We show that the BA.1 NTD amplifies the immune evasion of its associated RBD. BA.1 NTD reduces neutralistion by RBD directed monoclonal antibodies, and impacts ACE2 interaction. NTD variation is thus an important mechanism of immune evasion by SARS-CoV-2. Such effects are not seen when pre-stabilized spike proteins are used, suggesting the interdomain effects require protein mobility to express their phenotype.

## Author summary

SARS-CoV-2 continues to evolve into new variants bearing RBD and NTD mutations in spike that confer immune escape. Phenotypic maps of SARS-CoV-2 spike mutations used in surveillance and forecasting are based largely on data using screens with monomeric RBD, rather than trimeric whole spike. Interactions between the NTD and RBD affect receptor binding affinity and immune escape of the SARS-CoV-2 spike. Using deep

scores can be found in the supplementary excel file: "Summary_enrichment_files_.xlsx". All the raw sequence data will be deposited in the Short Read Archive https://www.ncbi.nlm.nih.gov/sra/PRJNA962104.

**Funding:** This work received funds from Wellcome (https://wellcome.org/), Wellcome fellowship no. 216353/Z/19/Z (RK, WB), BRC through Imperial College Healthcare NHS Trust (https://imperialbrc.nihr.ac.uk/) RDA01 (WB), and the MRC (https://www.ukri.org/councils/mrc/) MR/W005611/1 (WB). The funders had no role in study design, data collection and analysis, decision to publish, or preparation of the manuscript. RK received a salary from Wellcome fellowship no. 216353/Z/19/Z. RK, KS, and MM received a salary from Imperial College Healthcare NHS Trust- BRC Funding RDA01.

**Competing interests:** The authors have declared that no competing interests exist.

mutational scanning with whole trimeric spike, we identified antibody escape mutations that have subsequently emerged in nature. Vaccine responses were found to be focused on one or two epitopes, making immune escape possible with single mutations. Because we used whole spike, we uncovered an unexpected role for the NTD in enhancing antibody escape from RBD directed antibodies. Genomic surveillance should take account of this new role of the NTD when scanning for VOCs.

## Introduction

SARS-CoV-2 variants with increased immune escape and transmissibility have emerged repeatedly with mutations in spike [1]. The SARS-CoV-2 spike exists as a trimer in its native state. The S1 subunit contains an N-terminal domain (NTD) and the receptor binding domain (RBD) that interacts with the ACE2 receptor. The S2 subunit contains the fusion machinery [2]. S1/S2 cleavage is carried out by host cell proteases and a further S2' cleavage event primes the spike for fusion. Phenotypic maps to predict mutations in the SARS-CoV-2 spike that will increase ACE2 binding and immune evasion have been produced using deep mutagenesis scanning [3–9] and directed evolution [10]. However, the majority of these studies have been performed using just the RBD, which when expressed by itself is monomeric. Whilst evolution studies using monomeric RBD have successfully predicted the emergence of key RBD mutations, the maps might be improved by using whole SARS-CoV-2 spike in its native trimeric form in which both intra and inter-domain epistasis can play out. Increasingly studies with deep mutagenesis using whole spike are contributing to the knowledge base of mutational effects. These have included mammalian cell displays of SARS-CoV-2 spike to explore the effect of mutations on the NTD [11,12] and using pseudovirus systems with whole spike deep mutational scanning (DMS) libraries [9]. Natural evolution reveals multiple mutations in both the RBD and NTD of variants of concern, VOCs (Fig 1). The exact role of the NTD is uncertain, yet deletions, insertions and substitutions are strongly selected for in the NTD of SARS-CoV-2 variants [1,13]. Mutations of the NTD have been shown to alter spike processivity

| | 19 | 24 | 25 | 26 | 27 | 67 | 69 | 70 | 95 | 142 | 143 | 144 | 145 | 156 | 157 | 158 | 211 | 212 | 213 | 214 |
|---|---|---|---|---|---|---|---|---|---|---|---|---|---|---|---|---|---|---|---|---|
| Wuhan | T | L | P | P | A | A | H | V | T | G | V | Y | Y | E | F | R | N | L | V | R |
| Alpha | T | L | P | P | A | A | - | - | T | G | V | - | Y | E | F | R | N | L | V | R |
| Delta | R | L | P | P | A | A | H | V | T | D | V | Y | Y | G | - | - | N | L | V | R |
| BA.1 | T | L | P | P | A | A | - | - | I | D | - | - | - | E | F | R | I | - | V | INS |
| BA.2 | I | - | - | - | S | V | H | V | T | D | V | Y | Y | E | F | R | N | L | G | R |
| BA.2.12.1 | I | - | - | - | S | V | H | V | T | D | V | Y | Y | E | F | R | N | L | G | R |
| BA.4 / BA.5 | I | - | - | - | S | V | H | V | T | D | V | Y | Y | E | F | R | N | L | G | R |

| | 339 | 371 | 373 | 375 | 376 | 405 | 408 | 417 | 440 | 446 | 452 | 477 | 478 | 484 | 486 | 493 | 496 | 498 | 501 | 505 |
|---|---|---|---|---|---|---|---|---|---|---|---|---|---|---|---|---|---|---|---|---|
| Wuhan | G | S | S | S | T | D | R | K | N | G | L | S | T | E | F | Q | G | Q | N | Y |
| Alpha | G | S | S | S | T | D | R | K | N | G | L | S | T | E | F | Q | G | Q | Y | Y |
| Delta | G | S | S | S | T | D | R | K | N | G | R | S | K | E | F | Q | G | Q | N | Y |
| BA.1 | D | L | P | F | T | D | R | N | K | S | L | N | K | A | F | R | S | R | Y | H |
| BA.2 | D | F | P | F | A | N | S | N | K | G | L | N | K | A | F | R | G | R | Y | H |
| BA.2.12.1 | D | F | P | F | A | N | S | N | K | G | Q | N | K | A | F | R | G | R | Y | H |
| BA.4 / BA.5 | D | F | P | F | A | N | S | N | K | G | R | N | K | A | V | Q | G | R | Y | H |

**Fig 1. Alignment of NTD and RBD of SARS-CoV-2 variants.** Top) NTD alignments, Bottom) RBD alignments of Wuhan, Alpha, Delta, BA.1, BA.2, BA.2.12.1 and BA.4/BA.5. Shading of cells represents the variant in which the mutation occurred first.

[14,15], cell-cell fusion [16] and have been suggested to have a role in antibody evasion [17,18].

The ongoing evolution of SARS-CoV-2 spike has largely selected for two phenotypes: increased ACE2 binding and antibody evasion [19]. RBD-directed antibodies account for ~90% of the neutralising antibody response of convalescent [7] and vaccine sera [8], while neutralising NTD antibodies contribute a minority [18]. Nonetheless, NTD mutants emerge from chronic infections in immunocompromised hosts [20–23] and in a directed evolution experiment selecting for SARS-CoV-2 escape mutants using convalescent sera [24]. Thus, forward evolution studies using a monomeric RBD in isolation may miss important effects on ACE2 binding and antibody evasion mediated by other domains of the SARS-CoV-2 spike.

Here, we developed a whole trimeric spike mammalian cell display platform that uses deep mutational scanning to identify mutations that increase ACE2 binding or lead to immune evasion. We used the DMS platform with an Alpha spike to show that the E484K or Q498R RBD mutations increase ACE2 binding. The effect of Q498R on ACE2 binding was dependent on the presence of the N501Y RBD mutation. This combination of Q498R and N501Y is present in all currently circulating Omicron subvariants [25]. We also used the whole trimeric spike platform to show how post-vaccine immune responses are focused on one or two epitopes on the SARS-CoV-2 spike RBD, and that single mutations at these immunodominant epitopes account for most of the escape seen with Omicron VOC BA.1 spike, despite its 15 mutations in the RBD [26]. Unexpectedly, the amount of escape seen with RBD mutations did not account for the total escape of BA.1 spike from vaccine sera. Using chimeric SARS-CoV-2 spikes with NTD domain swaps, we show the BA.1 NTD enhances the antibody evasion by RBD and that this inter-domain epistasis underlies the high level of immune escape seen with the Omicron variants.

## Results

### E484K and Q498R mutations enhance binding of Alpha VOC spike to human ACE2

At the time of experimental conception, Alpha was the dominant variant and hence was chosen as the base spike for this study. To explore the potential evolutionary space of the Alpha spike-RBD, a library was made using degenerate primers (NNK) and overlapping PCR, on an Alpha spike base sequence, tagged at the C-terminal end with a fluorescent protein (mGreen-Lantern [27]) and cloned into a mammalian expression plasmid (pcDNA3.1). The library covered 4183 of 4220 (99.1%) of possible single mutations, present as predominantly single or double mutations across the 211 amino acids residues in RBD (S1 Fig). Transfected cells displaying spike at their surface were probed with a soluble human ACE2-Fc protein tagged with mScarlet [28]. Spike expressing cells with the highest ACE2 binding were sorted using FACS and sequenced using next-generation sequencing (NGS) and the proportions of each variant compared with the library (Fig 2A). We focused our analysis to the RBD residues that were in the interface with ACE2 receptor, and also those that are altered in Omicron BA.1 Spike. The two most enriched substitutions that increased ACE2 binding of Alpha spike were E484K and Q498R (Fig 2B).

To confirm the effect on ACE2 binding in our whole spike expression platform, Alpha spike mutants with E484K or Q498R were constructed and binding to ACE2 measured using flow cytometry. ACE2 binding was corrected for spike expression as shown in Fig 2C. The introduction of either E484K or Q498R increased ACE2 binding relative to wild type (WT) Alpha-spike (Fig 2D and 2E). We note that E484K did not appear to have as large an effect on ACE2 binding in a yeast DMS scan using an RBD containing N501Y [29]. To confirm the

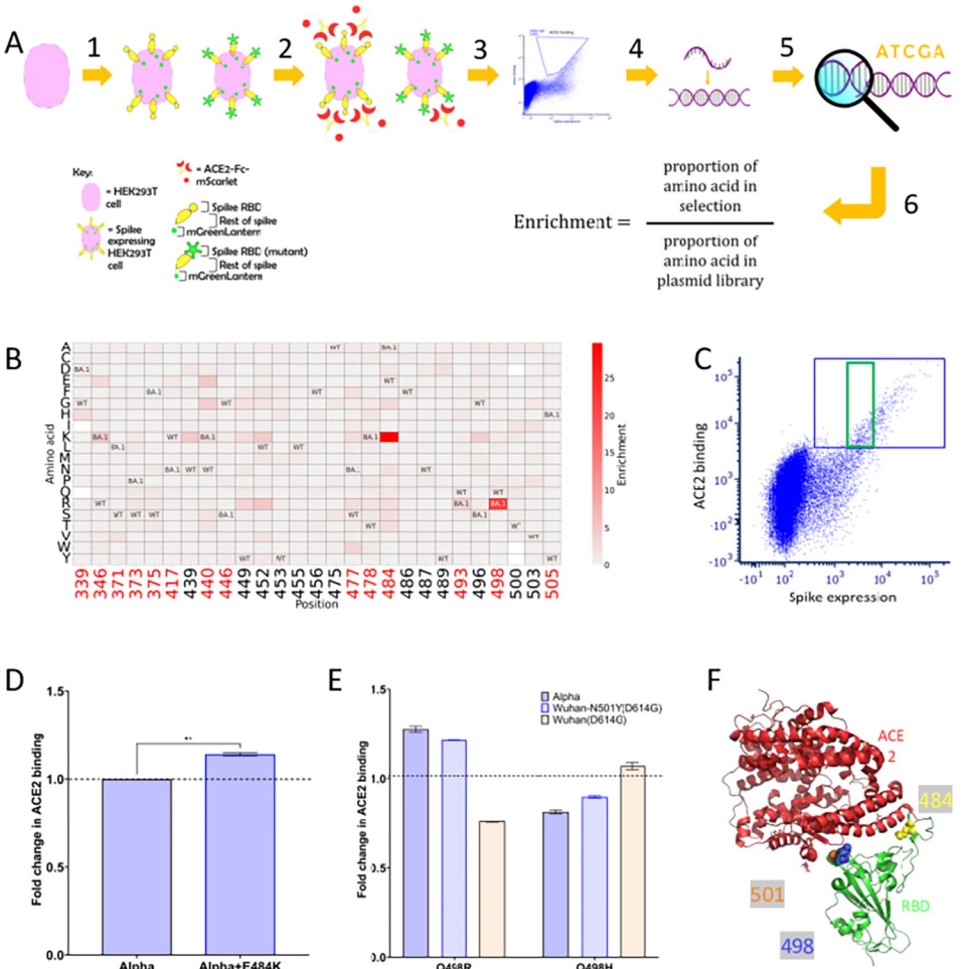

**Fig 2. Prediction and validation of the effect of mutations of the Alpha-RBD and the role of epistasis on ACE2 binding.** (A) Schematic of experimental design for enriching spike mutants with high ACE2 binding (see Methods for further detail). (B) Heatmap of point mutations showing their enrichment score for ACE2 binding. The RBD positions shown have been filtered to those involved in the ACE2 binding interface [3] and mutations that occurred in Omicron VOC BA.1 spike [26]. Red numbers on the x axis, represent positions mutated in BA.1. WT = wild type amino acid. BA.1 = amino acid found in BA.1. Blank squares represent point mutations not present in the library. (C) HEK-293T cells were transfected with the trimeric spike tagged at the C-terminal end with mGreenLantern. 24 hours later, cells were incubated with ACE2-Fc-mScarlet for 1 hour. Spike expression is shown on the X axis and ACE2 binding on the Y axis. The green box shows ACE2 binding corrected for spike expression, while the blue box shows total binding from the ACE2 positive cell population. Relative ACE2 binding for (D) E484K on Alpha, (E) Q498R and Q498H on Alpha, Wuhan+N501Y(D614G) and Wuhan(D614G) trimeric spike. Data presented is the fold difference in median ACE2 binding for each mutant relative to the parent spike corrected for expression (green box Fig 1C). N = 2, error bars represent the range. ** p value < 0.01, one-way Anova. (F) RBD positions 484(yellow), 498(blue), 501(orange) are directly involved in the interaction with hACE2(red). PDB: 6M0J [54]. Fig 1F created using PyMOL (The PyMOL Molecular Graphics System, Version 2.0 Schrödinger, LLC.).

increase in binding we were observing in the context of whole Alpha Spike, we measured ACE2 binding across a range of different concentrations of purified recombinant ACE2 and confirmed the ACE2 binding increase to the E484K mutation at each of these concentrations (S2 Fig).

Previous work using DMS on a Wuhan-spike RBD identified Q498H but not Q498R as a substitution that increased ACE2 binding [3]. A DMS and directed evolution study showed that the effect of mutation at 498 was impacted by residue 501 [10,29,30]. Indeed, the Wuhan

and Alpha RBD differ by the presence of the N501Y mutation in Alpha. To further verify this difference, the Q498R or Q498H mutations were introduced to the Alpha, Wuhan+N501Y (D614G) and Wuhan(D614G)-spikes and their ACE2 binding measured (Fig 2E). In the context of a spike with N501Y-RBD, either Alpha or Wuhan based, Q498R increased ACE2 binding, whereas Q498H reduced it. By contrast in spike with a 501N-RBD, the addition of Q498H led to an increase in ACE2 binding, whereas Q498R reduced it.

Positions 498 and 501 are in close proximity to each other on the RBD (Fig 2F). Structural studies illustrate the incompatibility of N501Y and Q498H would be due to steric clashes between the aromatic residues [10].

## Antigenic escape by mutations at amino acid 477 is detected using whole spike DMS

We next used the Alpha spike library to screen for mutations that escape the monoclonal antibodies (mAbs) LyCoV-016, REGN10987, and REGN10933 (Fig 3A, 3B and 3C). Escape mutations against these mAbs have been well documented using a Wuhan RBD [5,31], allowing us to explore if epistasis conferred by mutations in spike changes the profile of escape mutations. Cells expressing spike variants that retained the greatest ability to bind ACE2 in the presence of the mAb were sorted and processed as described above.

The escape maps generated for the screened mAbs largely agreed with those from Wuhan-RBD based DMS screens [5,31]. A sample of predicted escape mutations were independently verified by pseudovirus neutralisation assays (Fig 3A, 3B and 3C). However, using the full-length spike platform, residue 477 appeared as a site of escape from REGN10933 (Fig 3C), a site not enriched in monomeric Wuhan-RBD screens [5,31]. To explore if this was due to epistasis or differing methodologies, pseudovirus bearing the mutations S477D and S477P in a Wuhan(D614G) or an Alpha spike were constructed and neutralisation assays against REGN10933 conducted. The S477D and S477P mutations were chosen as they were predicted to have the largest effects at position 477 (Fig 3D). Fig 3D shows the S477D and S477P mutations caused similar decreases in neutralisation regardless of spike background. The structure of spike with REGN10933 mAb reveals this residue sits in the antibody epitope (Fig 3E).

## DMS reveals most polyclonal vaccine sera select for mutations at one or two antigenic sites in spike RBD

Having validated the platform's ability to identify mAb escape mutants, we next screened the library for escape from polyclonal sera. Blood was collected from 8 healthy adults between 2–4 weeks after their 2nd BNT162b2 vaccine dose (S1 Table).

As might be expected, the mutations that enabled escape from different vaccine sera were heterogeneous, although residues at 484 and 452 were the most frequently selected, and some of the most enriched (Fig 4A and 4B).

To better visualise the mutations that enable escape from most sera across the cohort, the data from the eight individual vaccine sera escape maps were combined. Fig 4C shows the combined frequency of escape, ie how many of the substitutions at that position led to escape, for each position in the RBD, highlighting that vaccine-induced antibodies predominantly target the immunodominant site at residue 484 and also to a lesser extent the second site at position 452.

Antigenic evolution leading to fixation of amino acid substitutions would be expected to occur in a direction that escapes the dominant immune focus seen in most people, even if a particular unique mutation exerts a strong escape for some individuals. To identify which amino acid substitutions would exert the most dramatic escape in the double vaccinated

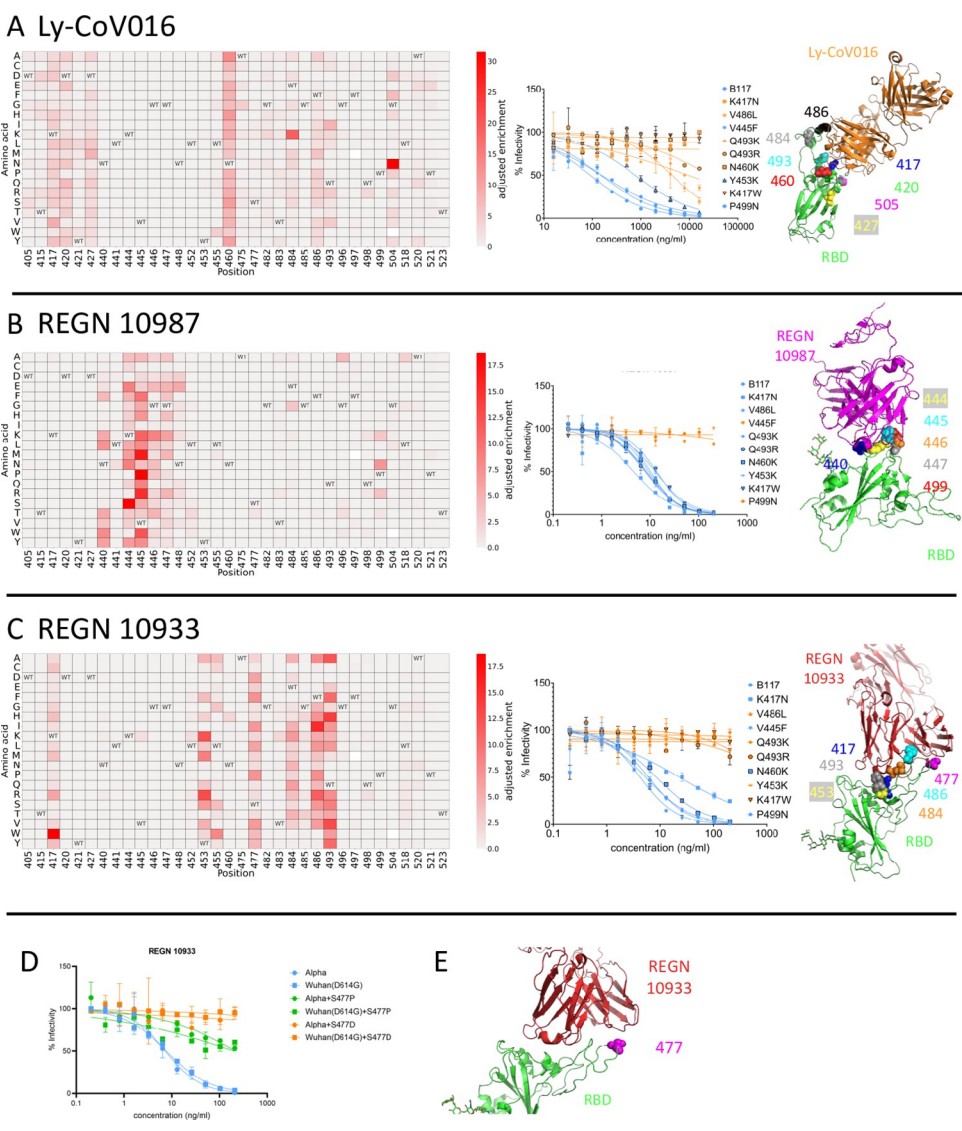

**Fig 3. mAb escape heatmaps for (A) Ly-CoV016, (B) REGN 10987 and (C) REGN 10933.** To the immediate right of each heatmap are plots showing pseudovirus neutralisation curves for a selected panel of pseudovirus mutants. To the furthest right are structural representations of the respective mAb binding to the RBD, with predicted positions of escape highlighted. The RBD positions presented in the heatmap are those where escape occurs most frequently in the mAbs. WT = wild type amino acid. Blank squares represent amino acids not present in the library. Adjusted enrichment is the enrichment score multiplied by the fraction of amino acids other than WT that would be predicted to escape. (D) Pseudovirus neutralisation curves for point mutations S477D and S477P on an Alpha or Wuhan (D614G) pseudovirus compared to WT Alpha and Wuhan(D614G) pseudovirus. All neutralisation assays were performed in duplicate. (E) Structural representation of REGN 10933 binding to SARS-CoV-2 RBD. PDB: 6XDG [55]. HIV based pseudovirus platform with hACE2 over-expressing HEK-293T cells used for neutralisation assays.

cohort, the adjusted enrichment ratios for each RBD amino acid were summated across the cohort and are presented as a combined heatmap with a number in each cell to represent how many individual sera selected this substitution for escape (Fig 4D). The substitutions with the highest enrichment scores and affecting the highest proportion of sera were predominantly located on the ACE2 binding face of the RBD (445, 452, 483, 484, 490, 493). The positively charged amino acids, arginine and lysine in this region showed the greatest enrichment.

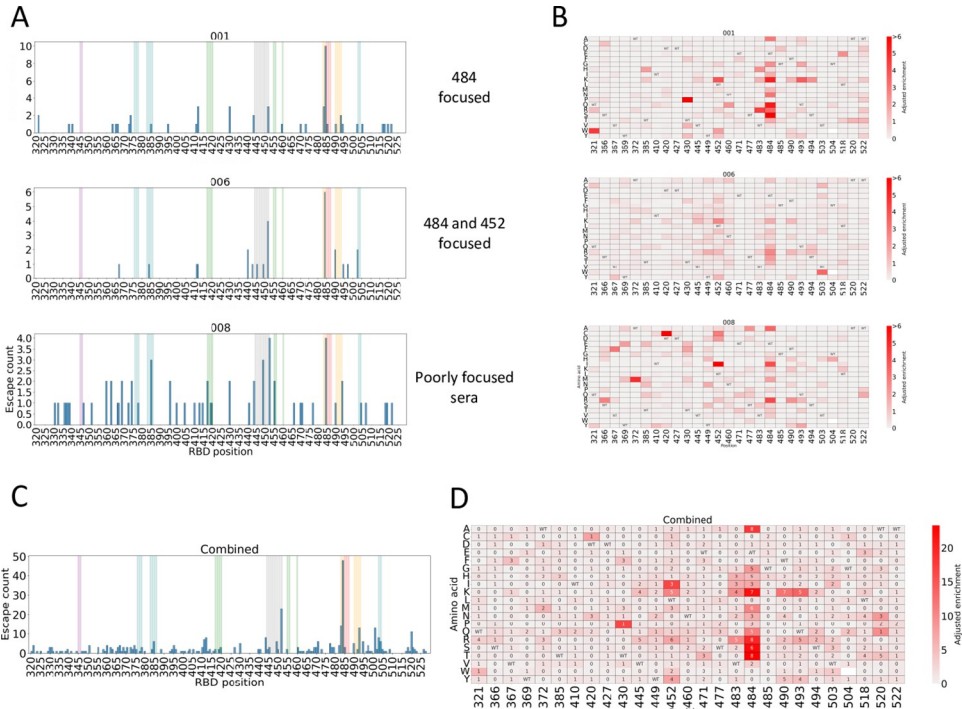

**Fig 4. Double BNT162b2 vaccine immune responses are heterogenous and focused on a limited number of positions on the RBD.** (A) Histograms representing the frequency an amino acid at each position in the RBD escapes vaccine sera. Bars represent the number of amino acids at each RBD position that have an adjusted enrichment score >1. Individual histograms are shown for a sample of sera. The histograms are representative of patterns seen across all the sera. (B) Escape maps for a sample of the vaccine sera. (C). Cumulative escape histogram from 8 vaccine sera. Bars represent the number of amino acids at each RBD position that have an adjusted enrichment score >1, the maximum possible being 19 (19 other amino acids than WT) for an individual sera x 8 (summated across all 8 sera) = 152. The coloured bars represent classes of mAbs [31]: Class A = Green, Class B = Red, Class C = Orange, Class D = Gray, Class E = Purple, Class F = Blue. (D). Cumulative escape heatmap from summating adjusted enrichment scores from 8 vaccine sera. Adjusted enrichment is the enrichment score multiplied by the fraction of amino acids other than WT that would be predicted to escape. RBD positions shown are those having the highest frequency of amino acids across all the sera with adjusted enrichment scores >1. WT = wild type, blank squares = amino acids not represented in the library. The number in each cell represents the number sera that the mutation had an adjusted enrichment score greater than 0.5.

Position 484 has the most amino acids enriched for escape in at least half of tested sera, followed by position 452, 490 and 493 (Fig 4D).

To explore the effect of the predicted escape mutations on vaccine neutralisation, a selection of the most enriched mutations were engineered into Alpha spike and pseudovirus neutralisation assays conducted. Single mutations in Alpha spike RBD had a modest effect on escape, ranging from a 1.3-fold decrease for E484K to a 1.9-fold decrease for Q493K. The combination of 3 mutations, L452R, E484K and Q493K only led to a 2.4-fold decrease in neutralisation from vaccine sera (Fig 5A).

## Two RBD mutations in Omicron BA.1 (E484A and Q493R) account for the large antigenic escape by the BA.1 RBD

BA.1 emerged in November 2021 as the most antigenically distant SARS-CoV-2 variant known at that time. The BA.1 spike contains 15 mutations in the RBD relative to Wuhan-RBD including N501Y seen in Alpha and the E484A and Q493R mutations highlighted by our DMS (Fig 5B). Pseudovirus with BA.1 spike showed a mean 9-fold decrease in neutralization titre by

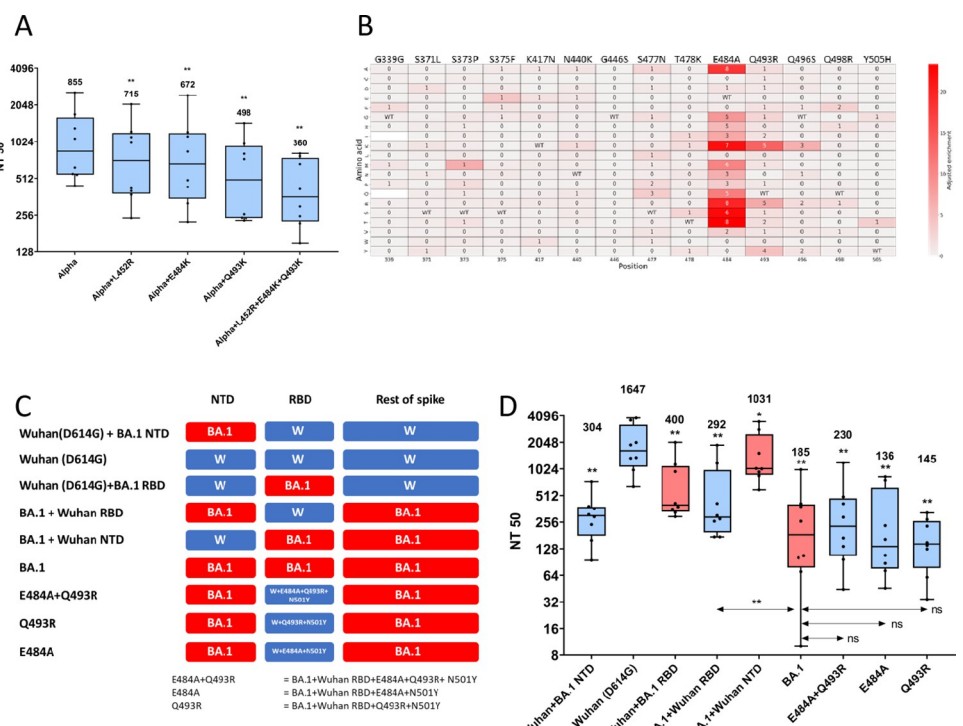

**Fig 5. The antigenic escape of the BA.1-RBD from double dose BNT162b2 vaccine sera can be accounted for by single mutations and is enhanced by the BA.1 NTD.** (A) Pseudovirus neutralisation assays using the vaccine sera against Alpha spike and Alpha spike mutants to identify the effect of single mutations on escape. (B) The cumulative double dose BNT162b2 vaccine escape map filtered to the positions mutated in BA.1 relative to Alpha to show the contributions of each mutation in BA.1 to vaccine escape. WT = wild type, blank squares = amino acids not represented in the library. The number in each cell represents the number sera that the mutation had an adjusted enrichment score greater than 0.5. (C) Schematic of chimeric pseudoviruses constructed and used in neutralisation assays. W = Wuhan(D614G). (D) Pseudovirus neutralisation assays using 8 vaccine sera against pseudovirus bearing WT spike, mutant and chimeric spikes identify their respective contributions to vaccine escape, as shown in the schematic. Shading of the box plots represents the parent RBD, Wuhan = blue, BA.1 = red. Median NT50 is shown.

our vaccine sera cohort (Fig 5D). To confirm whether the combination of 15 mutations in the RBD are responsible for the large escape seen, compared with the relatively small antigenic distance (2.4-fold mean titre drop) measured by the combination of 3 mutations at 452, 484 and 493 (Fig 5A), a chimeric spike was generated using Wuhan(D614G) spike with the RBD replaced with the BA.1 RBD (Fig 5C). Surprisingly, neutralisation assays using this chimeric spike only led to a 4-fold decrease in neutralisation by the polyclonal vaccine sera. Interestingly, the converse chimeric construct, full BA.1 spike with the RBD replaced by Wuhan-RBD led to a greater escape, a 5.6-fold decrease in neutralisation (Fig 5D). The escape heatmap predicts that, of the 14 RBD mutations in BA.1 RBD relative to Alpha, E484A and Q493R would contribute the most to BA.1-RBD's vaccine escape (Fig 5B). To test their importance, we created a BA.1 chimeric spike protein with a Wuhan RBD containing E484A, Q493R and N501Y (E498A + Q493R in Fig 5C). The neutralization titre for the 8 vaccine sera was not significantly different against this construct than against full BA.1. In fact, the presence of either of E484A or Q493R alone with N501Y in the context of the BA.1 full spike did not show any significant change in neutralisation from WT BA.1, suggesting the effects of E484A and Q493R are not additive and the BA.1 RBD has a redundancy with respect to mutations leading to immune escape.

## Mutations in the Omicron BA.1 NTD have a large impact on the neutralization of the RBD

The negligible effects of most BA.1 RBD mutations and unexpected effects of domains outside of the RBD on the extent of BA.1 neutralisation prompted us to look more deeply into the impact of the NTD on immune escape. Chimeric pseudoviruses featuring NTD swaps between BA.1 and Wuhan(D614G) spike were constructed and the effect on vaccine neutralisation assessed (Fig 5D). Changing the NTD of Wuhan(D614G)-spike to BA.1-NTD reduced neutralisation by the vaccine sera by 5.6-fold, while replacing the BA.1-NTD with Wuhan NTD made the chimeric spike 5.6-fold easier to neutralise than the parental BA.1 spike (Fig 5D and S2 Table). Summing up data from the various chimeric constructs presented in Fig 5C and 5D, the BA.1-NTD reduced the antibody neutralisation of any spike it was part of. The same effect was seen using similar chimeric spike proteins with mixture of domains from Wuhan and Delta (S3 Fig).

The effect seen with the NTD chimeras is surprising given that most of the neutralising activity in polyclonal sera has been ascribed to RBD-directed antibodies. To exclude the possibility the unexpected effects of BA.1 NTD on neutralization were due to the sample of vaccine sera having predominantly NTD-focused immune responses, chimeric pseudoviruses containing the Wuhan-RBD with BA.1 NTD were used in neutralisation assays with mAbs targeting the Wuhan-RBD. We reasoned that, if the surrounding domains have no effect on RBD recognition, then the mAb $IC_{50}$ should be independent of domains other than the RBD. In the presence of the BA.1-NTD, the Wuhan-RBD was 2.3 to 5-fold less readily neutralized by Ly-CoV016, REGN10987, and REGN10933 (Fig 6A, 6B, 6C and S3 Table). The NTD has been shown to affect spike cleavage and entry efficiency [14] which could affect the interpretation of effects on neutralization. To exclude the influence of this in our neutralisation assays, pseudovirus input had already been normalised by infectious units. In addition, to assess whether there was differential processing or incorporation of spike into pseudoviruses, a Western blot was performed on a subset of the pseudoviruses bearing chimeric spikes with NTDs exchanged (Figs 6D and S4). There was no difference in the efficiency of spike cleavage or incorporation between Wuhan(D614G) and BA.1+Wuhan RBD or Wuhan+BA.1 NTD that could explain the neutralisation differences (Figs 6D and S4). Thus, the BA.1-NTD appears to make the virus harder to neutralise by altering its recognition by antibodies targeted to the RBD.

These findings contrasted with the findings from Javanmardi et al. [32] who showed little effect of BA.1 NTD domain on the binding of RBD directed mAbs. We hypothesized the different findings may be related to the use of pre-fusion stabilized SARS-CoV-2 spikes by Javanmardi et al. who engineered 6 proline substitutions in S2 to increase spike expression [32,33]. The effect we see of the BA.1 NTD on RBD directed mAbs may require a native S2 to transmit a dynamic interaction between the BA.1 NTD and neighbouring RBD. To assess this hypothesis, we engineered matched chimeric spike proteins with domains from BA.1 and Wuhan spike with and without pre-fusion stabilization in S2 and compared binding of RBD-directed mAbs using flow cytometry (Fig 6E, 6F and 6G). We found that pre-fusion stabilization abrogated the effect of the BA.1 NTD on mAb binding to the RBD in agreement with Javanmardi et al [32] (Fig 6E, 6F and 6G). However, in the non-stabilized spike, the BA.1 NTD significantly reduced RBD-directed mAb binding in agreement with our pseudovirus neutralization data. This implies that the interdomain effects between the NTD and RBD that impact antibody binding and likely also on ACE2 binding require the S2 to be mobile (Fig 6E, 6F and 6G).

To further explore NTD epistasis on the RBD, the effect on ACE2 binding was measured. The BA.1 spike binds ACE2 20% better than Wuhan (D614G) spike. Replacing the RBD of BA.1 with Wuhan RBD returns ACE2 binding back to the level of Wuhan (D614G).

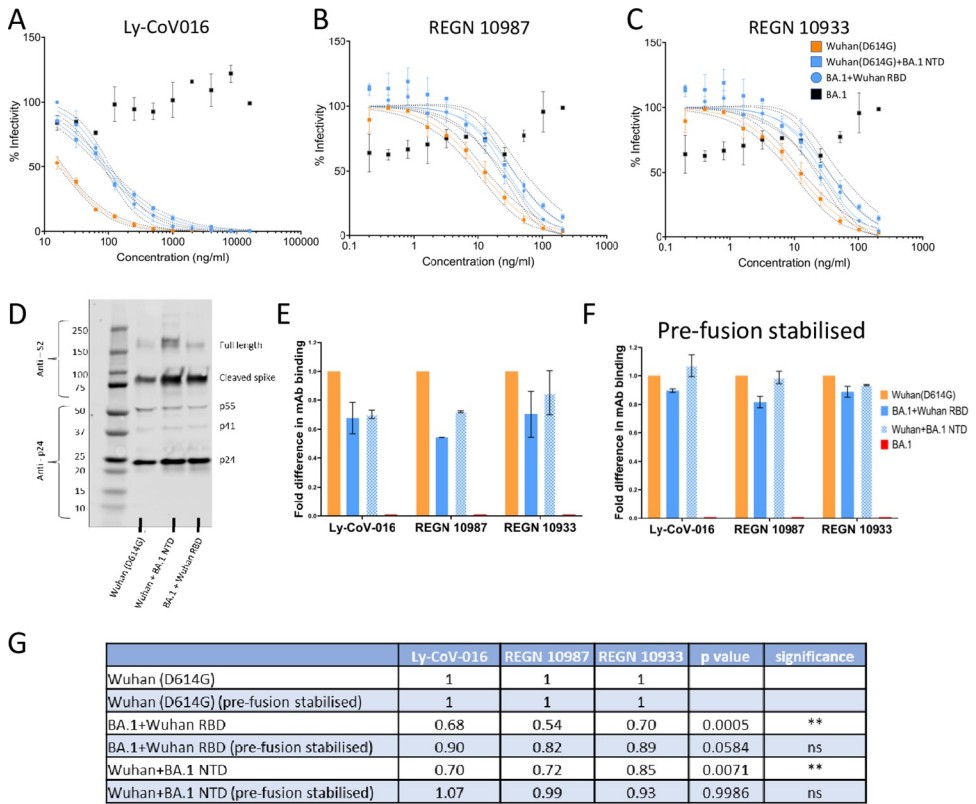

**Fig 6. Reduction in antibody binding to the SARS-CoV-2 RBD by the BA.1 NTD depends on Spike protein mobility.** Neutralisation assays of pseudovirus with spike containing Wuhan-RBD with different NTD by mAbs targeting SARS-CoV-2-RBD. (A) Ly-CoV016, (B) REGN10987, (C) REGN10933. Dashed lines around each fitted curve represent the 95% confidence interval. *p value <0.05, ** p value <0.01, significantly different from BA.1 (Wilcoxon matched-pairs sign rank test). ns = non–significant. (D) Cleavage and spike incorporation into pseudoviruses measured using Western blot. (E) Binding of Ly-CoV-016, REGN 10987, and REGN 10933 to SARS-CoV-2 spike chimeric proteins displayed on HEK-293T cells measured using flow cytometry. The fold change in median fluorescence intensity compared to Wuhan (D614G) is shown. (F) Binding of Ly-CoV-016 REGN 10987 and REGN 10933 by SARS-CoV-2 spike chimeric proteins that have been pre-fusion stabilised using 6 proline substitutions [33] displayed on HEK-293T cells measured using flow cytometry. The fold change in median fluorescence intensity compared to the pre-fusion stabilised Wuhan (D614G) spike is shown. (G) Table showing the mean fold difference in antibody binding by SARS-CoV-2 spikes with a Wuhan RBD and different combinations of BA.1 and Wuhan NTD and S2. The p values represent the significance of the difference in binding from their respective parent spike either Wuhan(D614G) or Wuhan(D614G) pre-fusion stabilised across all three mAbs tested collectively (Tukey's multiple comparisons). ** p value <0.01, ns = non-significant. HIV based pseudovirus platform with hACE2 over-expressing HEK-293T cells used for neutralisation assays. All neutralisation assays and binding assays were performed in duplicate.

Surprisingly, replacing the NTD and S2 domain of BA.1 with Wuhan equivalents reduced the ACE2 binding of spike by a similar level (S5 Fig). The replacement of Wuhan spike domains with any of BA.1 NTD, RBD and S2 domains increased ACE2 binding to a similar level to BA.1 whole spike. This highlights the importance of domains outside of the RBD in determining ACE2 binding of whole spike and underscores the advantages of working with full length proteins (S5 Fig).

## Discussion

Most phenotypic maps of the SARS-CoV-2 RBD have relied on expression of deep mutated libraries based on monomeric RBD [3–8]. This approach, although simpler than the one taken

here may, overlook the impact of other domains of spike, even on those phenotypes directly attributed to RBD if there is significant interdomain interaction. Here, we used DMS with trimeric whole spike displayed on human cells. Using Alpha spike as the base, we identify two mutations in the RBD, E484K and Q498R, that increase ACE2 binding. E484K was predicted to cause a small increase in ACE2 binding in a yeast DMS [29], and fixed early in a directed evolution study of the RBD after N501Y looking for a high affinity RBD using rounds of error-prone PCR [30]. Biolayer interferometry measurements have confirmed the addition of E484K to a N501Y RBD does increases the affinity to ACE2 [30]. In addition, 2 independent studies using surface plasmon resonance to measure RBD binding to ACE2 also showed E484K increases ACE2 binding both in the presence and absence of N501Y [34,35]. We show the effect of Q498R is epistatically linked to N501Y, in agreement with other published studies [10,29,30]. Q498R is a key mutation in the Omicron lineages that increases ACE2 binding and compensates for antigenic escape mutations that may be deleterious to binding [29]. The detection of epistasis highlights that such maps will require updating as SARS-CoV-2 spike continues to evolve. We validated our method of DMS to identify escape variants from monoclonal antibodies. These were largely in concordance with work from yeast DMS [5,31], however using our method we identified position 477 as a site of escape from REGN 10933 that was not described in yeast RBD DMS. Structural data confirms residue 477 is in the binding footprint of REGN 10933 and neutralisation assays confirm its role in antigenicity. Thus, while yeast RBD DMS mostly agrees with our model, using the more physiological trimeric spike and mammalian cells may be particularly important for identifying epitopes at interprotomeric regions [36]. In addition, screening using ACE2 competition to select for escape variants, rather than for reduced antibody binding, provides an in-built measure of the balance between escape and ACE2 binding and weights towards fitter escape mutants that are more likely to emerge as highly transmissible variants, as well as providing a better proxy for neutralisation escape.

Using double BNT162b2 vaccine sera, we showed vaccine responses to be highly focused on 2 epitopes encompassing residues 484 and 452. The surprisingly focused nature of the polyvalent immune response against the RBD has been shown before in convalescent [7] and vaccine sera [8]. Mutations at positions 484 and 452 have appeared recurrently in nature in variants associated with immune escape, Beta (E484K) [37], Delta (L452R) [38], BA.1 (E484A) [26], and BA.4/BA.5(L452R & E484A) [39]. Here, we aggregated the escape mutations from our vaccine sera cohort to predict mutations that may be important for escape in future variants. We predicted that mutations at the ACE2 binding face, in particular residues 452, 484, 490 and 493 would be the most important for mediating immune escape (Fig 4D). In the BA.1 spike, despite the accumulation of 15 amino acids changes in the RBD, we unexpectedly found that a single RBD mutation of E484A or Q493R accounts for most of the escape seen from vaccine sera. The presence of both mutations was not additive, and in nature Omicron variants have since appeared that revert the Q493R mutation, unsurprisingly given its redundancy and deleterious effect on ACE2 binding [40]. In the dominant Omicron lineages that emerged after BA.1, including the most recent BQ1.1 and XBB variants, convergent evolution at RBD positions (452, 490) that we identified here has been reported [40].

In this study, we selected for spike mutations using sera from vaccinees after 2 doses. The immune experience that drives forward evolution is now much more complex, including hybrid immunity and quadruple-boosted vaccine regimens. Nonetheless, our results may remain relevant due to the phenomenon of imprinting [40–43]. Most of the world have had their first immunising event with a Wuhan-based spike either through infection or vaccination. Immune imprinting, where antibodies that cross react against the first immune exposure are preferentially boosted has been recognised in influenza immunity [44] and now with

SARS-CoV-2 spike immunity [40–43]. Consequently, despite the heterogeneity in immune experience, most people have antibodies that recognize the Wuhan spike most avidly. Thus, we might expect to continue to observe the sequential and cumulative selection of mutations that escape the Wuhan antibody response, until variants arise that are sufficiently antigenically distant to stimulate novel and specific antibody responses. This concept should be tested in the future by performing DMS selection with updated sera cohorts from individuals with a variety of immune experience of SARS- CoV-2.

The emergence of BA.1, the herald of the Omicron lineages, was the most significant step in antigenic distance [45]. Antibodies against the RBD of earlier VOCs were shown to account for ~90% of neutralisation from convalescent or vaccine sera [7,8]. However, we show here that the BA.1 NTD was necessary for the large antigenic distance seen with BA.1 and this was not due to a higher proportion or increased potency of antibodies directed against NTD itself, but rather due to an inter-domain epistasis effect of NTD on recognition of RBD epitopes. The BA.1-NTD, when coupled to any RBD, made that RBD more difficult to neutralize. We even saw the same effect with mAbs against the RBD and impacts of NTD on ACE2 binding that is determined by direct interaction between RBD and the receptor. This effect of the BA.1 NTD on antibody binding of the RBD was only present with a native S2 domain and not when the spike was pre-fusion stabilised, suggesting that protein mobility is required to transmit the impact from one domain to the next. In nature, the most antigenically distant variants have repeatedly had NTD changes [1,13] and directed evolution experiments for immune escape from convalescent sera have seen NTD mutations coupled with a single RBD mutation for complete escape [24].

This study has limitations. The DMS screens were performed with only a single biological replicate and the low frequency cut-off used in analysis of the sequencing data introduces noise into the data, particularly for the variants present at low frequencies in the library. Additionally for these low frequency variants, possible enrichment may be missed due to the limits of detection of the sequencing methodology. However, we have validated the results of the most enriched mutants, to eliminate them being false positives, and accept the presence of other enriched mutations in the dataset that have not been validated could be the result of noise. In addition, it is possible that differing levels of spike expression of different spike mutants or chimeras might impact the ACE2 binding measurements we report. We attempted to mitigate this by measuring ACE2 binding over a window of unsaturated spike expression covering a dynamic range, which was monitored using the mGreenlantern tag on Spike protein. We did not find a large difference in Spike expression that could explain the differences in ACE2 binding we report.

This study uses deep mutagenesis scanning with whole trimeric spike displayed on mammalian cells to screen for mutations that increase ACE2 binding and escape antibodies. We show the importance of epistasis in interpreting genotype to phenotype data, the focused nature of immune responses on the RBD and the importance of the NTD in immune escape. Surveillance strategies should monitor for NTD changes, in addition to RBD changes, as novel NTDs have the potential to change RBD phenotype.

## Materials and methods

### Plasmids

The codon-optimized SARS-CoV-2 spike (Wuhan) in pcDNA3.1 was a gift from P. McKay [46], Imperial College London. Site directed mutagenesis using the QuikChange Lightning Site-Directed Mutagenesis Kit (210518) was used to introduce the spike mutations of Alpha. The BA.1 spike plasmids were a kind gift from T.Peacock. To generate the mutant spike

plasmids and domain swap chimeric spike plasmids a combination of site directed mutagenesis and DNA assembly was used (NEB #E2621).

The C-terminal end of the Alpha spike sequence was tagged with mGreenLantern [27] subcloned from pcDNA3.1-mGreenLantern, which was a gift from Gregory Petsko (Addgene plasmid # 161912; http://n2t.net/addgene:161912; RRID:Addgene_161912) using a glycine-serine rich linker (GSGGSGSGG) [47] to replace the last 21 amino acids of the spike sequence.

pmScarlet_C1 [28] was a gift from Dorus Gadella (Addgene plasmid # 85042; http://n2t.net/addgene:85042; RRID:Addgene_85042)

pcDNA3-sACE2(WT)-Fc(IgG1) [47] was a gift from Erik Procko (Addgene plasmid # 145163; http://n2t.net/addgene:145163; RRID:Addgene_145163)

mScarlet was subcloned from pmScarlet_C1 and tagged to the end of the Fc tag of sACE2(WT)-Fc(IgG1) to create sACE2(WT)-Fc(IgG1)-mScarlet.

## Serum samples

Serum samples were collected from 8 healthy adults aged 29–50, at least 2 weeks following and within 1 month of the second dose of the BNT162b2 mRNA vaccine (S1 Table). The serum samples were stored at −20°C at the local University Communicable Disease Research Tissue Bank (NRES SC/20/0226).

## RBD library construction

Mutagenesis primers containing the degenerate codon NNK were made for each amino acid in the RBD of a codon optimised SARS-CoV-2 spike using python script [48,49] available from https://github.com/jbloomlab/CodonTilingPrimers. The mutagenesis primers were used in a single round of overlap extension PCR consisting of 10 cycles of mutagenesis PCR, followed by 20 cycles of joining PCR. The PCR libraries were then cloned into a pcDNA3.1 plasmid containing Alpha spike tagged at the C-terminus with mGreenLantern using NEBuilder HiFi DNA Assembly (NEB #E2621). The assembled products were used to transform NEB DH5α ultracompetent cells (NEB # C2987P), following a 1-hour period of outgrowth at 37°C, the outgrowth media and transformed cells were poured into liquid LB media containing ampicillin and cultured overnight at 37°C. Plasmids were extracted and purified using the Qiagen HiSpeed Plasmid Maxi kit (12663).

Details on the average library coverage and proportions of mutations in the library can be found in supplementary Figs 1 and 6.

## FACS sorting

1ng of the plasmid library was transfected with 2000ng of empty plasmid per 10^6 of HEK 293T cells using Lipofectamine 3000 (Thermofisher L3000001). Co-transfection with an empty plasmid reduces the number of the coding plasmids that are transfected into a single cell [36,47,50,51] and allows resolution of genotype from phenotype.

24 hours later, the HEK-293T cells are dissociated and incubated with monoclonal antibodies or vaccine sera for 30 minutes. The cells are then washed in ice cold PBS with 5% FBS (FACS buffer) before being incubated with sACE2-Fc-mScarlet for a further 30 minutes. The cells are washed 2 times with FACS buffer and analysed on a BD FACS Aria III. Cells were initially gated for single cells and dead cells excluded using DAPI (1/2000). Sort populations were gated based on mScarlet and mGreenLantern signals (S7 Fig). Sorted cells were collected in an Eppendorf containing FACS buffer, the sorted cells were spun down at 100g for 10 minutes, supernatant removed, and the remaining cell pellets stored at -80C.

For ACE2 binding sorts the Alpha spike expressing HEK-293T cells were incubated with sACE2-Fc-mScarlet at a subsaturating dilution determined by titration on flow cytometry for 30 minutes without prior sera/mAb incubation (S8 Fig), then processed as above.

For the antibody escape sorts, a concentration of mAb or dilution of sera was chosen following titration with the Alpha spike RBD library that reduced ACE2 binding of the total population of cells by approximately 50%. Alpha spike expressing HEK-293T cells were initially incubated with the mAb or sera for 30 minutes, followed by washing with FACS buffer and then another incubation with a saturating volume of sACE2-Fc-mScarlet supernatant (S8 Fig). The concentrations of mAbs used for each sort are as follows: Ly-CoV-016 (400 ng/mL), REGN 10933 (80 ng/mL, REGN 10987 (160 ng/mL), while the dilutions of each sera used can be found in S1 Table.

Batches of 11x10^6 HEK-293T cells were stained, incubated, and washed at a time. Approximately 5–10% of these cells expressed the SARS-CoV-2 spike library due to the method of transfection used. From this total spike expressing cell population, the top 10% of ACE2 binding spike expressing HEK-293T cells were sorted until at least 10,000 cells were collected. The whole plasmid library contains over 4000 mutants, which are represented by at least 5*10^5 spike expressing cells from a total of 11*10^6 cells transfected cells, the sorted 10,000 cells will contain only a fraction of the mutants in the original plasmid library. The sorted 10,000 cells will contain predominantly mutants conferring the beneficial phenotype being selected for in that particular screen. From such a diverse library only a handful of mutations will be positively selected for and be represented multiple times over in the sorted population of at least 10,000 cells. If less than 10,000 target cells were collected in a single sort, further sorts were then conducted until 10,000 target cells were collected. The target population of cells were frozen at -80˚C and pooled with cells from a subsequent sort if required. The sorts were done in batches of about 2 hours, as over time the ACE2 signal would drop and have to be re-gated during the sort due possibly to ACE2 dissociation from spike or spike shedding of S1 [36]. The sorted cells from the batched sorts were then pooled together for RNA extraction.

### sACE2-Fc(IgG)-mScarlet

The sACE2-Fc(IgG)-mScarlet plasmid was transfected into HEK-293T cells at a ratio of 1000ng per 10^6 cells. 48 hours later the supernatant was harvested and filtered through a 0.45μm filter and stored in aliquots at -200C. An aliquot was titrated for binding to spike by flow cytometry prior to use.

### RNA extraction and sequencing

Total RNA was extracted from sorted cells using the Qiagen RNAeasy mini kit (74104) and reversed transcribed using SuperScript IV (Thermofisher 18090050) with gene specific primers. The RBD was amplified as 2 amplicons in the first round of PCR. A further round was PCR was used to add on Nextera XT indices (Illumina fc-131-1001) for barcoding and sequencing on the Illumina Miseq with 300bp paired end reads.

### ACE2 binding

Spike expressing cells were incubated with the ACE2-IgG(Fc)-mScarlet supernatant for 30 minutes, before being washed with FACS buffer and binding of ACE2 being measured using flow cytometry.

Spike constructs were tagged with mGreenLantern at the C-terminal end, allowing expression of spike by cells to be measured using the signal from mGreenLantern. ACE2 binding was measured using the median fluorescence intensity signal from the ACE2-IgG(Fc)-mScarlet.

ACE2 binding was measured from a population of cells with the same level of spike expression to avoid differences in spike expression caused by a mutation affecting measurements of ACE2 binding. Spike expression was gated to the lowest level of spike expression with detectable ACE2 binding and extended for half a log, while remaining on the linear portion of the ACE2 binding–spike expression plot. An example of this is shown by the green box in the dot plot in Fig 1C.

## Lentiviral pseudotype neutralisation assays

A codon optimised SARS-CoV-2 spike plasmid (pcDNA3.1) was a kind gift from Paul McKay. A stop codon was introduced in the C terminal tail of spike to delete the last 19 amino acids of the cytoplasmic tail and further point mutations to generate specific variants were created by site directed mutagenesis.

The HIV1 gag-pol (pCMV-Δ8.91) and firefly luciferase reporter (pCSFLW), plasmids used to produce the lentiviral pseudotypes were a gift from Paul MacKay [46].

To generate lentiviral particles pseudotyped with spike; spike plasmid, HIV1 gag-pol and a firefly luciferase reporter plasmid were transfected in the ratio of 1:1:1.5 into HEK-293T cells using Lipofectamine 3000. 72 hours later supernatant containing the pseudotyped lentiviruses was harvested, passed through a 0.45µm filter and stored at -80C.

The pseudotyped lentiviruses were titrated by serial dilution in pre-seeded 96 well plates of ACE2 over expressing-HEK-293T cells. After 48 hours of incubation, RLU (relative luciferase units) was measured using the Bright-Glo Luciferase Assay System (Promega).

For neutralisation assays, sera or monoclonal antibodies were serially diluted in a 96 well plate and ~$5*10^5$ RLU of pseudotyped lentivirus was added to the dilutions. The virus-antibody mix was incubated at 37oC for 1 hour, before the addition ACE2-HEK-293T cells and incubated at 37˚C for 60–72 hours. The RLU was measured using the Bright-Glo Luciferase Assay System (Promega). The NT(neutralisation titre)50% was calculated using GraphPad Prism by fitting the data to a Hill curve with GraphPad Prism (version 9.2.0) [52].

## Western blots

Pseudoviruses were concentrated by ultracentrifugation before being reduced in Laemmlli buffer with 10% β-mercaptoethanol. Proteins were transferred to a nitrocellulose membrane and were probed overnight with a polyclonal rabbit anti-SARS spike protein (NOVUS; NB100-56578) and mouse anti-p24 (abcam; ab9071), followed by 1 hour incubation with the secondary antibodies, IRDye 680RD Goat anti-mouse (abcam; ab216776) and IRDye 800CW Goat anti-rabbit (abcam; ab216773). Fluorescence was measured using the Odyssey Imaging System (LI-COR Biosciences).

## Data analysis

Raw fastq files were filtered and trimmed using Biopython 1.79 [53]. Trimmed sequences were aligned and translated using Geneious Prime 2019.2.1. A frequency cut-off of 0.001% was used in determining the proportions of variants in the library, then values differing by greater than 4-fold between two independent sequencing runs of the original plasmid library were excluded to reduce the likelihood these were due to sequencing error. To further reduce the effect of noise, mutations at the ACE2 binding face of the RBD were focused on as mutations having a positive effect on ACE2 binding would be expected to interact with ACE2. For the antibody escape screens, to reduce noise, an adjusted enrichment score for antibody escape was used. RBD positions important in antibody escape will generally have multiple mutations at this

position capable of causing escape, so greater weighting was given to mutations occurring at positions with multiple other enriched mutations (see adjusted enrichment equation below).

The raw enrichment scores can be found in the supplementary excel file: "Summary_enrichment_files_.xlsx". All raw sequencing data can be found at: https://www.ncbi.nlm.nih.gov/sra/PRJNA962104.

Enrichment scores were calculated by using the formula below:

$$Enrichment = \frac{proportion\ of\ amino\ acid\ in\ selection}{proportion\ of\ amino\ acid\ in\ plasmid\ library}$$

Adjusted enrichment scores used in monoclonal antibody and vaccine sera screens were calculated using the formula below:

$$Adjusted\ enrichment = \frac{proportion\ of\ amino\ acid\ in\ selection}{proportion\ of\ amino\ acid\ in\ plasmid\ library} \times \frac{number\ of\ amino\ aicds\ at\ this\ position\ with\ enrichment\ score > 2}{19\ (number\ of\ other\ possible\ amino\ acids)}$$

## Supporting information

**S1 Fig. Diversity of Alpha-RBD library.** Distribution of number of mutations per RBD. Following transformation with plasmid containing whole Alpha spike with the RBD library, 50 colonies, were picked, mini prepped and Sanger sequenced across the RBD, the number of mutations in each RBD is shown in the histogram above.
(TIF)

**S2 Fig. The E484K mutation increases Alpha SARS-CoV-2 spike binding to ACE2.** HEK-293T cells were transfected with Alpha and Alpha+E484K plasmids. 24 hours later cells were dissociated and incubated with recombinant hACE2-Fc(IgG) (Abcam ab273885) for 30 minutes. The cells were then washed and incubated with a secondary antibody against human Fc (IgG) (Abcam ab98622) for 30 minutes. ACE2 binding was measured as the median fluorescence intensity using flow cytometry. ACE2 binding was measured across a range of concentrations of recombinant hACE2-Fc(IgG) (Abcam ab273885), with all measurements being conducted in duplicate. The table shows the fold increase in ACE2 binding by Alpha+E484K relative to Alpha.
(TIF)

**S3 Fig. The Delta NTD accounts for the escape seen from BNT162b2 vaccine sera.** Pseudovirus neutralisation assays using double dose BNT162b2 vaccine sera against chimeric spikes involving domain swaps between Delta spike and Wuhan(D614G) spike. Shown are the median fold differences in NT50 from Wuhan(D614G) pseudovirus. HIV based pseudovirus platform with hACE2 over-expressing HEK-293T cells used for neutralisation assays.
(TIF)

**S4 Fig. Quantification of Western blot from Fig 5D measuring spike cleavage and incorporation into pseudovirus.** All quantification is normalised against Wuhan (D614G). (A) Proportion of cleaved to total spike. (B) Total spike per pseudovirus. Error bars represent standard error of the mean. N = 2.
(TIF)

**S5 Fig. Domains outside of the RBD are important for ACE2 binding.** ACE2 binding was measured by flow cytometry. HEK-293T cells were transfected with the respective plasmid, 24 hours later cells were dissociated and incubated with sACE2-Fc-IgG-mScarlet for 1 hour, before measuring the median fluorescence intensity. The mfi was corrected to spike expression in the same way as described in Fig 1. Shown is the relative difference in mfi to Wuhan (D614G). n = 2. ** p value < 0.001 using one-way ANOVA relative to Wuhan(D614G). (TIF)

**S6 Fig. Correlation between 2 independent sequencing runs of the plasmid library.** The RBD of the Alpha plasmid library was sequenced independently in duplicate by NGS. The proportion of reads containing each point mutation was calculated. The axes represent the proportion of reads for each point mutation in each of the 2 sequencing runs. Point mutations with proportions differing by a greater than 4-fold difference between replicates were excluded from the library. The scatter plot shows the correlation between proportions of point mutations included in the subsequent analyses. R value (Pearson rank coefficient). (TIF)

**S7 Fig. FACS gating strategy for high ACE2 binding and mAb/sera escape.** a. FSC-A against SSC-A to remove dead cells and debris. b. FSC-A against FSC-H to remove doublets. c. FSC-A against 405-450/50-A to remove dead cells by staining with DAPI. D. 488-525/50-A (spike expression) against 561-582/15 –A (ACE2) to select spike expressing cells. e. 488-525/50-A against 561-582/15-A gate for ACE2 positive spike expressing cells. f. 488-525/50-A against 561-582/15-A, the top 10% of ACE2 binding cells are sorted from an ACE2 positive, spike expressing population. The spike protein used were all tagged at the C-terminal end with mGreenLantern and the ACE2-IgG Fc tagged with mScarlet. (TIF)

**S8 Fig. sACE2-Fc-mScarlet titrations for FACS.** HEK-293T cells were transfected with the sACE2-Fc-mScarlet plasmid at a ratio of 1000ng per 10^6 cells. After 48 hours the supernatant was collected, filtered and aliquoted for storage at -20˚C. sACE2-Fc-mScarlet from this batch was used to incubate ~5*10^5 HEK-293T cells transfected 24 hours previously with plasmid expressing Alpha SARS-CoV-2 spike tagged with mGreenLantern for 30 minutes. The Alpha SARS-CoV-2 spike tagged with mGreenLantern plasmid was transfected using 1ng of the spike expressing plasmid with 2000ng of empty plasmid per 10^6 HEK-293T cells. ACE2 binding is shown as the median fluorescence intensity (mFI) of mScarlet following incubation with a range of volumes of the sACE2-Fc-mScarlet supernatant. (TIF)

**S1 Table. Participant demographics.** The table provides dates of vaccinations with BNT162b2, bleeds, ages and gender of the people who donated sera used in this study. The dilution of sera used to incubate with the Alpha spike library is shown. (XLSX)

**S2 Table. The BA.1-NTD reduces the neutralization of BNT162b2 vaccine sera.** Pseudovirus neutralization assays using chimeric spike bearing pseudovirus involving domain swaps between BA.1 spike and Wuhan (D614G) spike against vaccine sera. Shown are the median fold differences in NT50 between the parent spikes and the chimeric spikes. Green shaded represents a decrease in neutralization, red shaded represents increased neutralization. (XLSX)

**S3 Table. The BA.1-NTD reduces the neutralization of RBD directed mAbs.** Pseudovirus neutralisation assays using Ly-CoV016, REGN 10987 and REGN 10933 mAbs against chimeric

spikes involving domain swaps between BA.1 spike and Wuhan(D614G) spike. Shown are the median fold differences in NT50 from Wuhan(D614G) pseudovirus.
(XLSX)

## Acknowledgments

"We thank the St. Mary's NHLI FACS core facility and their staff in particular Radhika Patel for support and instrumentation"

"The Imperial BRC Genomics Facility has provided resources and support that have contributed to the research results reported within this paper. The Imperial BRC Genomics Facility is supported by NIHR funding to the Imperial Biomedical Research Centre".

"We thank Anne Palser of Kymab for providing the monoclonal antibodies used in this study."

## Author Contributions

**Conceptualization:** Ruthiran Kugathasan, Paul Kellam, Wendy Barclay.

**Data curation:** Ruthiran Kugathasan.

**Formal analysis:** Ruthiran Kugathasan.

**Funding acquisition:** Wendy Barclay.

**Investigation:** Ruthiran Kugathasan, Ksenia Sukhova, Maya Moshe.

**Methodology:** Ruthiran Kugathasan, Ksenia Sukhova, Maya Moshe.

**Resources:** Paul Kellam.

**Supervision:** Wendy Barclay.

**Validation:** Ruthiran Kugathasan.

**Visualization:** Ruthiran Kugathasan.

**Writing – original draft:** Ruthiran Kugathasan, Paul Kellam, Wendy Barclay.

**Writing – review & editing:** Ruthiran Kugathasan, Paul Kellam, Wendy Barclay.

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
