## [Decision Letter · Decision Letter 0]

2 Mar 2023

Dear Dr Kugathasan,

Thank you very much for submitting your manuscript "Deep mutagenesis scanning using whole trimeric SARS-CoV2 spike highlights the importance of NTD-RBD interactions in determining spike phenotype." for consideration at PLOS Pathogens. As with all papers reviewed by the journal, your manuscript was reviewed by members of the editorial board and by several independent reviewers. In light of the reviews (below this email), we would like to invite the resubmission of a significantly-revised version that takes into account the reviewers' comments.

Thanks for submitting your manuscript. It has been reviewed by two expert reviewers. Both agree it is interesting, but suggest substantial revisions. Essentially all of the reviewer comments seem good to me. Please pay special attention to the following ones:

- Raw data need to be available. Probably FASTQ on SRA, but definitely CSV files giving all of the mutation-level measurements plotted in the various heat maps.

- Clarifying how spike expression was corrected for.

- Use a diverging color scale for the heatmaps (https://clauswilke.com/dataviz/color-basics.html#color-to-represent-data-values). The current scale is not intuitive or easy to understand.

- Better methods descriptions of various points noted by reviewers, including data analysis.

- Better citations to prior literature performing mammalian cell display or full spike deep mutational scanning. Also some quantitative comparison with results of such studies could be helpful, particularly if you cannot address the comment from Reviewer 1 about replicates of your own experiments.

In addition, I would add the following point not raised by the reviewers: the very large increase in ACE2 affinity due to E484K is surprising. Many prior studies have looked at affinity enhancing mutations, and have not generally found E484K to have a large effect even in a N501Y background. Given the surprising nature of this result, some further validation should be performed using an orthogonal assay, or at least flow cytometry at titrated levels of ACE2.

We cannot make any decision about publication until we have seen the revised manuscript and your response to the reviewers' comments. Your revised manuscript is also likely to be sent to reviewers for further evaluation.

Sincerely,

Jesse D Bloom, Ph.D.

Guest Editor

PLOS Pathogens

Sara Cherry

Section Editor

PLOS Pathogens

Kasturi Haldar

Editor-in-Chief

PLOS Pathogens

orcid.org/0000-0001-5065-158X

Michael Malim

Editor-in-Chief

PLOS Pathogens

orcid.org/0000-0002-7699-2064

Thanks for submitting your manuscript. It has been reviewed by two expert reviewers. Both agree it is interesting, but suggest substantial revisions. Essentially all of the reviewer comments seem good to me. Please pay special attention to the following ones:

- Raw data need to be available. Probably FASTQ on SRA, but definitely CSV files giving all of the mutation-level measurements plotted in the various heat maps.

- Clarifying how spike expression was corrected for.

- Use a diverging color scale for the heatmaps (https://clauswilke.com/dataviz/color-basics.html#color-to-represent-data-values). The current scale is not intuitive or easy to understand.

- Better methods descriptions of various points noted by reviewers, including data analysis.

- Better citations to prior literature performing mammalian cell display or full spike deep mutational scanning. Also some quantitative comparison with results of such studies could be helpful, particularly if you cannot address the comment from Reviewer 1 about replicates of your own experiments.

In addition, I would add the following point not raised by the reviewers: the very large increase in ACE2 affinity due to E484K is surprising. Many prior studies have looked at affinity enhancing mutations, and have not generally found E484K to have a large effect even in a N501Y background. Given the surprising nature of this result, some further validation should be performed using an orthogonal assay, or at least flow cytometry at titrated levels of ACE2.

Reviewer's Responses to Questions

**Part I - Summary**

Reviewer #1: Kugathasan et al. carried out deep mutational scanning using whole trimeric spike protein to characterize the effect of alpha RBD mutations on ACE2 binding as well as antibody escapes. The results not only captured many known observations, but also contain new findings (e.g. mutations at position 477 can strongly escape REGN10933). The authors also performed a domain swapping experiment and showed that NTD has an important effect to both ACE2 binding and the RBD antibody escape. Although this manuscript offers potentially interesting biological insights, many essential information and analyses are missing from the manuscript. Thus, the rigor and effectiveness of the screen cannot be evaluated.

Reviewer #2: Kugathasan and colleagues present a mammalian cell display platform for high-throughput characterization of mutations in the SARS-CoV-2 spike. They confirm ACE2 affinity-enhancing RBD mutations (and epistatic interactions) previously described in isolated RBD studies, and identify antibody-escape mutations in full spike context that were not strongly escaping in isolated RBD studies. Last, the authors suggest that Omicron NTD changes have as much or more of an immune-escape effect compared to RBD changes, a striking finding that leads to my primary critique below. Overall, I think this is an important study that needs just a bit more interpretation for the surprising results seen with NTD chimeras.

**Part II – Major Issues: Key Experiments Required for Acceptance**

Reviewer #1: 1. Reproducibility of the DMS was not assessed. How many independent biological replicates has been done for the screening? What is the correlation between replicates? These were not mentioned throughout the manuscript.

2. The effect of RBD mutations on ACE2 binding have been measured by two previous DMS studies (PMID: 33597251 and PMID: 32841599). What is the correlation between the data in this present study with those two previous studies?

3. Figure 4 E,F,G: Neutralization assays on BA.1 should also be included as a control.

4. Page 8: “The enhancement of ACE2 binding by the BA.1-NTD was not conferred unless the BA.1-RBD was coupled with its cognate NTD (Fig S4).” This claim needs to be either modified or supported by including Wuhan spike + BA.1 RBD, Wuhan spike + BA.1 NTD, and Wuhan spike + BA.1 RBD + BA.1 NTD in the experiment, since it is possible that S2 domain also plays a role here. Currently, domain swapping in Fig S4 was only performed using BA.1 spike (i.e. BA.1 spike + Wuhan RBD and BA.1 spike + Wuhan NTD).

Reviewer #2: 1. Other citations with SARS-CoV-2 DMS platforms: PMID 36417523 and Cell doi 10.1016/j.cell.2023.02.001. Both are whole-spike platforms and probably should be mentioned alongside the methodology discussed here. Another important citation and experiments to contrast results with is PMID 35988543, see point below.

2. It is certainly unexpected as the authors say to see that there is a greater loss in serum neutralizing potency for the Wuhan-Hu-1 spike with the BA.1 NTD versus the BA.1 RBD (and its individual antibody-escape mutations). I do believe that NTD mutations can impact RBDs, but I do think the particular findings here warrant a more careful consideration. These results differ quite dramatically from the extensive chimera experiments performed by Javanmardi et al. (PMID 35988543 reference listed above), who used a mammalian spike display platform to extensively characterize BA.1 mutations and chimeras for mAb binding and other biochemical properties. Their Wuhan-1 + BA.1 NTD or BA.2 NTD constructs (Fig. 4A) do not impact binding by any of the RBD mAbs in their panel (including overlap with mAbs seen here), though these constructs for example do exhibit enhanced spike expression (consistent with Fig. 4H). Because Javanmardi et al. conduct binding assays which are more straightforward than neutralization assays, I think these results need to be carefully considered in light of contrasting results here. In particular, this comparison makes me wonder if something indirect is happening in the relationship between spike function/expression and pseudoviral infection and/or neutralization in the current manuscript that could give rise to artefactual or nonspecific changes in neutralizing potency due to indirect spike features and not antibody escape per se. For example, results from the present manuscript and Javanmardi et al. indicate that many of these spike chimeras can differ in spike stability/expression or ACE2 binding, and therefore presumably infectivity. The current study states that neuts were performed setting each experiment to an equal infectious titer. However, if that requires adding substantially more pseudoviral particles (or more directly, more spike protein) for some constructs versus others, then if serum antibodies or mAbs are not present in sufficient molar excess, couldn’t this systematically alter neutralizing IC50s by excess spike depletion of limiting mAb? I think the authors need to consider how much spike (functional or not) is present in different preps and consider whether that is systematically associated with their neutralization effects of chimeras. Regardless of this specific hypothesis I suggest here for the discrepencies in results, I think the authors need to carefully discuss their results in relationship to the Javanmardi et al. results in the present manuscript, as this is crucial to the interpretation of their data.

3. Data is only available upon request. At minimum, a table of all of the deep mutational scanning enrichments should be made available as supplementary data. Raw sequencing data is typically required to be uploaded to a repository, as well. I believe PLOS policy is that raw data for all plots should be available, too.

**Part III – Minor Issues: Editorial and Data Presentation Modifications**

Reviewer #1: 1. Throughout the manuscript, “SARS-CoV2” should be changed to “SARS-CoV-2”.

2. Deep mutational scanning of SARS-CoV-2 RBD using whole trimeric spike and mammalian display has been done and published before (PMID: 33597251). Moreover, antigenic escape screen using whole trimeric spike and mammalian display has also been done and published before (please see PMID: 34919820). However, the authors stated that they are the first to adopt such strategies. Therefore, the innovation part of this manuscript should be toned down.

3. Figure 1A: What is the green star in the scheme? An antibody, or a bead coated with antibodies?

4. Page 6: the meaning of “ACE2 binding normalised for spike expression” needs to be clarified. It is unclear what normalization procedure was performed here.

5. Figure 2: the color scale of the heat map is a bit confusing because there are two light colors at the extremes and dark colors in the middle.

6. Page 7: “E3484A” should be “E484A”.

7. Page 8: “E4984A” should be “E484A”.

8. Page 8: “(Fig 4D & table S3)” should only be “(Fig 4D)”.

9. In the methods section: “Co-transfection with an empty plasmid reduces the number of the coding plasmids that are transfected into a single cell [3, 6-8]”. I am not sure if the references are correct here because none of those studies seem to use this method.

10. In the methods section, the authors mentioned the possible shedding of S1 of spike during sorting. Did the author modify the furin cleavage site to minimize such issue?

11. What is the rationale of using the adjusted enrichment scores when analyzing antibody escape? It is unclear what is the point of including (number of amino acids at this position with enrichment score > 2)/(number of other possible amino acids) as a factor in the calculation of adjusted enrichment scores.

12. As shown in Figure S5, some mutations have extremely low frequency. Given that the error rate of Illumina sequencing typically ranges from 0.1% to 1%, it can be difficult to distinguish between real mutations and sequencing errors. As a result, the phenotypic measurement for low frequency variants in DMS can be very noisy. Did the authors apply a frequency cutoff to filter out low frequency variants?

13. What are the concentrations of sACE2-Fc-mScarlet and monoclonal antibodies being used in the DMS experiment?

14. In the methods section, the authors mentioned that “A target of ~10,000 cells were aimed to be sorted for each selection”. This is problematic considering the library size is more than 4,000. Sorting such low numbers of cells would not be sufficient to resolve the affinity hierarchy of all mutants. The authors have likely sorted a lot more cells than 10,000, but it was not mentioned in the manuscript and should be clearly stated.

15. Fig S5 and S6 are not cited in the main text.

16. The authors should deposit their raw sequencing data to a public database (e.g. Sequence Read Archive).

Reviewer #2: Because neutralization sensitivities to different classes of mAbs can differ depending on cell line and ACE2 expression levels (Farrell et al. Viruses 2022, Lempp et al. Nature 2021), it would be helpful in the figure legends on neutralization assays to state the pseudovirus platform and cell lines used.

PLOS authors have the option to publish the peer review history of their article (what does this mean?). If published, this will include your full peer review and any attached files.

Reviewer #1: No

Reviewer #2: No
---

## [Decision Letter · Decision Letter 1]

25 May 2023

Dear Dr Kugathasan,

Thank you very much for submitting your manuscript "Deep mutagenesis scanning using whole trimeric SARS-CoV-2 spike highlights the importance of NTD-RBD interactions in determining spike phenotype." for consideration at PLOS Pathogens. As with all papers reviewed by the journal, your manuscript was reviewed by members of the editorial board and by several independent reviewers. The reviewers appreciated the attention to an important topic. Based on the reviews, we are likely to accept this manuscript for publication, providing that you modify the manuscript according to the review recommendations.

Thanks for the revision. Although you have addressed some of the reviewer comments, both reviewers still highlight some points that are not adequately addressed. Please address all of these in another revision, and I will reconsider that. I'd like to highlight a few key points to pay special attention to:

- Reviewer 2 highlighted in the first round of reviews how a supplemental file with all of the measurements in CSV or Excel format was not available, and it appears that such a file is still not available. The measured effects of the mutations is the primary output of this experiment, and so a file with those direct numbers needs to be included with the paper. I will reject any further revisions that do not provide such a file.

- Reviewer 1 highlights a number of points about noise, lack of replicates, etc that seem valid. I concur with the reviewer that it is not necessary to fix all of these with new experiments at this point, but any further revision should clearly state and discuss these limitations so they are transparently described.

- Please also pay careful attention to the other reviewer comments.

Sincerely,

Jesse D Bloom, Ph.D.

Guest Editor

PLOS Pathogens

Sara Cherry

Section Editor

PLOS Pathogens

Kasturi Haldar

Editor-in-Chief

PLOS Pathogens

orcid.org/0000-0001-5065-158X

Michael Malim

Editor-in-Chief

PLOS Pathogens

orcid.org/0000-0002-7699-2064

Thanks for the revision. Although you have addressed some of the reviewer comments, both reviewers still highlight some points that are not adequately addressed. Please address all of these in another revision, and I will reconsider that. I'd like to highlight a few key points to pay special attention to:

- Reviewer 2 highlighted in the first round of reviews how a supplemental file with all of the measurements in CSV or Excel format was not available, and it appears that such a file is still not available. The measured effects of the mutations is the primary output of this experiment, and so a file with those direct numbers needs to be included with the paper. I will reject any further revisions that do not provide such a file.

- Reviewer 1 highlights a number of points about noise, lack of replicates, etc that seem valid. I concur with the reviewer that it is not necessary to fix all of these with new experiments at this point, but any further revision should clearly state and discuss these limitations so they are transparently described.

- Please also pay careful attention to the other reviewer comments.

Reviewer Comments (if any, and for reference):

Reviewer's Responses to Questions

**Part I - Summary**

Reviewer #1: The authors have addressed most my previous comments. However, several concerns remain to be fully addressed, including major ones.

Reviewer #2: provided in initial review

**Part II – Major Issues: Key Experiments Required for Acceptance**

Reviewer #1: 1. Pooling cells from independent cell sorts does not count as a biological replicate of the DMS. Without biological replicates of the DMS, the reproducibility of the DMS cannot be accessed This is a huge concern given that the data is likely to be very noisy (see major comments #3 and #4 below). While performing another biological replicates of the DMS will likely be very time consuming, the authors should at least acknowledge the caveat of having only one biological replicate of the DMS.

2. Line 248: “The BA.1 spike binds ACE2 20% better than Wuhan (D614G) spike. Replacing the RBD of BA.1 with Wuhan RBD returns ACE2 binding back to the level of Wuhan (D614G), surprisingly replacing the NTD and S2 domain of BA.1 with Wuhan equivalents reduce the ACE2 binding of spike by a similar level (Fig S5). The replacement of Wuhan spike domains with any of BA.1 NTD, RBD and S2 domains increases ACE2 binding to a similar level to BA.1 whole spike. This highlights the importance of domains outside of the RBD in determining ACE2 binding of whole spike (Fig S5).” It is possible that the domain swapping between the Wuhan (D614G) and the BA.1 spike leads to change in expression level of the spike chimera on the cell surface, therefore leading to the variation in ACE2 binding. Therefore, it is necessary to measure the spike expression level of these chimera constructs as a control.

3. Line 404: “Batches of 11x10^6 HEK-293T cells were stained, incubated, and washed at a time. Approximately 5-10% of these cells expressed the SARS-CoV-2 spike library due to the method of transfection used. From this total spike expressing cell population, the top 10% of ACE2 binding spike expressing HEK-293T cells were sorted until at least 10,000 cells were collected. If less than 10,000 target cells were collected in a single sort, further sorts were then conducted until 10,000 target cells were collected. The target population of cells were frozen at -80°C and pooled with cells from a subsequent sort if required.” This part is still quite confusing. It is true that as the authors stated, the transfection would generate enough cells to represent the whole library. However, if only 10,000 cells total within top 10% of the ACE2 binding spike expressing cells were collected, the results are likely to be very noisy given that the library contains >4,000 mutants.

4. Relatedly, lines 474: “A frequency cut-off of 0.001% was used in determining the proportions of variants in the library.” If a mutation has frequency of 0.001% (i.e. 1 in 100,000) and is enriched 5 times during selection, its frequency after selection will be 0.005% (1 in 20,000). However, such enrichment will unlikely be detected since only 10,000 cells were collected. Instead, there is a high chance such mutation will be absence in the collected cells and assigned an enrichment score of 0.

Reviewer #2: I appreciate the new discussion of the current results versus Javanmardi et al. added in revision. The newly added comparison of the NTD chimera results with or without prefusion stabilization is intriguing. This seems to be consistent with some of the observations on variant spike dynamics seen in e.g. Calvaresi et al. PMID 36918534. I think this and some of the other differences between the current data and prior studies are still not always 100% clear (biologically that is, not in the writing itself), but the present study discusses these differences and adds important new data to the overall understanding of spike dynamics and emergent quaternary features.

Line 198: “To test their importance, we created a BA.1 spike protein that retained these two mutations…” – I think the description of this would be better if it included in the description explicitly that this is a chimeric BA.1 spike where rest of the RBD besides these three substitutions is Wuhan-Hu-1 (if I understand correctly)

Line 114 and 264:

The cited yeast deep mutational scanning measurements for E484K report a consistent but small affinity-enhancing delta-log10Ka of between 0.11 and 0.24 in the reported backgrounds, corresponding to a 1.3- to 1.7-fold affinity increase, different than the no effect ascribed in the manuscript in these sections and in the reviewer response.

It is also incorrect in the reviewer response to discuss how the “relative size of the increase in ACE2 binding” as determined in the current assay quantitatively relates to the SPR/BLI measured affinities. The measurement here is very different – labeling at a single concentration with a dimeric ACE2 – from a true biochemical Kd as measured by BLI or SPR. Looking at the three cited studies, E484K has no effect on the Wuhan background in Zahradnik et al. in figure 2c (yeast display titration), and in Table 2 (BLI) there is no precise test of the affinity effect of E484K alone (it is simply tested in many sporadic combinatorial mutants, so its isolated effect is not clear). Barton et al. precisely find that E484K has a 1.19-fold enhancement affinity in Wuhan and 1.48-fold in N501Y background. Laffeber et al. measure E484K as having 1.4-fold affinity-enhancement in Wuhan RBD, and 1.7-fold on the N501Y background. Therefore, the BLI and SPR experiments cited, like the yeast DMS, ascribe a very small affinity-enhancing effect to E484K, at least in isolated RBD out of the spike context. It is entirely unclear how the “relative size” of increase identified here compares to these other studies because there is no absolute quantitative meaning to the deep mutational scanning enrichment or cell-surface expression labeling measurements as reported here. The only other reference is that here it is found to be on a similar order of magnitude with Q498R, whereas in various other studies Q498R (on top of N501Y) is seen to have a much larger effect on affinity than the <2-fold changes evident for E484K.

Zahradník, J., et al., SARS-CoV-2 variant prediction and antiviral drug design are enabled by RBD in vitro evolution. Nature Microbiology, 2021. 6(9): p. 1188-1198.

Barton, M.I., et al., Effects of common mutations in the SARS-CoV-2 Spike RBD and its ligand, the human ACE2 receptor on binding affinity and kinetics. eLife, 2021. 10: p. e70658.

Laffeber, C., et al., Experimental Evidence for Enhanced Receptor Binding by Rapidly Spreading SARS-CoV-2 Variants. Journal of Molecular Biology, 2021. 433(15): p. 167058.

Data availability: I would think a supplemental table csv file listing the final enrichment values from the deep mutational scanning data analysis would be a key inclusion in the manuscript

**Part III – Minor Issues: Editorial and Data Presentation Modifications**

Reviewer #1: 1. “Spike expression was gated to the lowest level of spike expression with detectable ACE2 binding and extended for half a log, while remaining on the linear portion of the ACE2 binding – spike expression plot.” I am still not entirely sure whether this procedure is well justified to correct for spike expression when measuring ACE2 binding. Half log (3-fold) difference in expression difference is still relatively large. It may be best to include a statement in the manuscript to acknowledge this caveat.

Reviewer #2: (No Response)

PLOS authors have the option to publish the peer review history of their article (what does this mean?). If published, this will include your full peer review and any attached files.

Reviewer #1: No

Reviewer #2: No

Figure Files:

Data Requirements:

Reproducibility:

References:

---

## [Editor Report · Decision Letter 2]

6 Jul 2023

Dear Dr Kugathasan,

We are pleased to inform you that your manuscript 'Deep mutagenesis scanning using whole trimeric SARS-CoV-2 spike highlights the importance of NTD-RBD interactions in determining spike phenotype.' has been provisionally accepted for publication in PLOS Pathogens.

Best regards,

Jesse D Bloom, Ph.D.

Guest Editor

PLOS Pathogens

Sara Cherry

Section Editor

PLOS Pathogens

Kasturi Haldar

Editor-in-Chief

PLOS Pathogens

orcid.org/0000-0001-5065-158X

Michael Malim

Editor-in-Chief

PLOS Pathogens

orcid.org/0000-0002-7699-2064

The authors have satisfactorily addressed the second round of reviewer comments, and I recommend for acceptance.
---

## [Editor Report · Acceptance letter]

25 Jul 2023

Dear Dr Kugathasan,

We are delighted to inform you that your manuscript, "Deep mutagenesis scanning using whole trimeric SARS-CoV-2 spike highlights the importance of NTD-RBD interactions in determining spike phenotype.," has been formally accepted for publication in PLOS Pathogens.

Best regards,

Kasturi Haldar

Editor-in-Chief

PLOS Pathogens

orcid.org/0000-0001-5065-158X

Michael Malim

Editor-in-Chief

PLOS Pathogens

orcid.org/0000-0002-7699-2064